# Maternal immunisation against Group B Streptococcus: A global analysis of health impact and cost-effectiveness

Simon R. Procter[1,2]*, Bronner P. Gonçalves[1,2], Proma Paul[1,2], Jaya Chandna[1,2], Farah Seedat[1,2], Artemis Koukounari[1,2], Raymond Hutubessy[3], Caroline Trotter[4], Joy E. Lawn[1,2], Mark Jit[1,5]*

1 Department of Infectious Disease Epidemiology, London School of Hygiene & Tropical Medicine, London, United Kingdom, 2 Maternal, Adolescent, Reproductive & Child Health (MARCH) Centre, London School of Hygiene & Tropical Medicine, London, United Kingdom, 3 Department of Immunization, Vaccines and Biologicals (IVB), World Health Organization, Geneva, Switzerland, 4 Disease Dynamics Unit, Department of Veterinary Medicine, University of Cambridge, Cambridge, United Kingdom, 5 School of Public Health, University of Hong Kong, Hong Kong SAR, China

* simon.procter@lshtm.ac.uk (SRP); mark.jit@lshtm.ac.uk (MJ)

**Data Availability Statement:** Code and data used in this analysis are available at https://github.com/mert0248/GBS-vax-econ-model.

**Funding:** SRP, BPG, PP, JC, FS, AK, JEL and MJ were supported by funding from a grant (INV-

## Abstract

### Background

Group B Streptococcus (GBS) can cause invasive disease (iGBS) in young infants, typically presenting as sepsis or meningitis, and is also associated with stillbirth and preterm birth. GBS vaccines are under development, but their potential health impact and cost-effectiveness have not been assessed globally.

### Methods and findings

We assessed the health impact and value (using net monetary benefit (NMB), which measures both health and economic effects of vaccination into monetary units) of GBS maternal vaccination in an annual cohort of 140 million pregnant women across 183 countries in 2020. Our analysis uses a decision tree model, incorporating risks of GBS-related health outcomes from an existing Bayesian disease burden model. We extrapolated country-specific GBS-related healthcare costs using data from a previous systematic review and calculated quality-adjusted life years (QALYs) lost due to infant mortality and long-term disability. We assumed 80% vaccine efficacy against iGBS and stillbirth, following the WHO Preferred Product Characteristics, and coverage based on the proportion of pregnant women receiving at least 4 antenatal visits. One dose was assumed to cost $50 in high-income countries, $15 in upper-middle income countries, and $3.50 in low−/lower-middle-income countries. We estimated NMB using alternative normative assumptions that may be adopted by policymakers.

Vaccinating pregnant women could avert 127,000 (95% uncertainty range 63,300 to 248,000) early-onset and 87,300 (38,100 to 209,000) late-onset infant iGBS cases, 31,100 deaths (14,400 to 66,400), 17,900 (6,380 to 49,900) cases of moderate and severe neurodevelopmental impairment, and 23,000 (10,000 to 56,400) stillbirths. A vaccine effective

009018) to the London School of Hygiene & Tropical Medicine (PI JEL) from the Bill & Melinda Gates Foundation. The funders had no role in study design, data collection and analysis, decision to publish, or preparation of the manuscript.

**Competing interests:** I have read the journal's policy and the authors of this manuscript have the following competing interests: FS is employed by the UK NSC which developed the policy recommendation for maternal GBS screening. RH is member of the World Health Organisation.

**Abbreviations:** ANC, antenatal care; CET, cost-effectiveness threshold; CFR, case fatality risk; EOGBS, early-onset GBS; FVVA, Full Value of Vaccines Assessment; GBS, Group B Streptococcus; GDP, Gross Domestic Product; HRQoL, health-related quality of life; IAP, intrapartum antibiotic prophylaxis; ICER, incremental cost-effectiveness ratio; iGBS, invasive GBS; LOGBS, late-onset GBS; NDI, neurodevelopmental impairment; NMB, net monetary benefit; PDVAC, Product Development for Vaccines Advisory Committee; PPC, preferred product characteristic; QALY, quality-adjusted life year; SDG, Sustainable Development Goal; UNWPP, United Nations World Populations Prospects; UR, uncertainty range; USD, United States Dollars; VE, vaccine efficacy.

against GBS-associated prematurity might also avert 185,000 (13,500 to 407,000) preterm births. Globally, a 1-dose vaccine programme could cost $1.7 billion but save $385 million in healthcare costs. Estimated global NMB ranged from $1.1 billion ($−0.2 to 3.8 billion) under the least favourable normative assumptions to $17 billion ($9.1 to 31 billion) under the most favourable normative assumptions.

The main limitation of our analysis was the scarcity of data to inform some of the model parameters such as those governing health-related quality of life and long-term costs from disability, and how these parameters may vary across country contexts.

## Conclusions

In this study, we found that maternal GBS vaccination could have a large impact on infant morbidity and mortality. Globally, a GBS maternal vaccine at reasonable prices is likely to be a cost-effective intervention.

## Author summary

### Why was this study done?

- Group B Streprococcus (GBS) is a common bacterial pathogen that can infect pregnant women and their babies.

- A recent global disease burden study showed that GBS infection causes a considerable burden of sepsis and meningitis in newborns, which can sometimes result in death or long-term disability, and it may also be linked to increased risk of stillbirth and preterm births.

- Several vaccines against GBS for use during pregnancy are being developed.

- A global economic evaluation of GBS vaccines is needed to inform investment decisions in vaccine development and to guide fair financing and pricing to enable equitable access once licensed vaccines become available.

### What did the researchers do and find?

- We developed a decision model to assess the cost-effectiveness of GBS vaccines in pregnant women in 183 countries for the year 2020.

- Our model used the most recent global estimates of the health burden of GBS in pregnant women and their children together with estimated costs to healthcare systems.

- We found that, globally, a maternal GBS vaccination programme, integrated in antenatal care, would lead to an overall increase in costs that are partially offset by savings in healthcare costs, along with substantial health gains, notably reductions in morbidity and mortality.

- Globally, the value of the annual GBS vaccine programme ranged from $1.1 billion (95% uncertainty range: $–0.2 to 3.8 billion) to $17 billion ($9.1 to 31 billion) depending on the normative assumptions used by policymakers.

### What do these findings mean?

- Globally, GBS maternal immunisation is likely to be cost-effective and avert a substantial burden of death and disability in children.

- At a regional and country level, cost-effectiveness is sensitive to vaccine prices and to different choices policymakers may use to value benefits in improved health.

- Our findings highlight the need both for carefully tiered vaccine pricing to ensure equitable access across countries and for local assessment of cost-effectiveness as GBS vaccine moves towards licensure.

- There is a need for more evidence on the impact of GBS on several outcomes, including stillbirths, preterm births, and maternal morbidity, as well as the wider societal costs of long-term GBS-related disability.

## Introduction

*Streptococcus agalactiae*, commonly known as Group B Streptococcus (GBS), is an important bacterial pathogen causing morbidity and mortality in pregnant women and their babies and is also increasingly recognised as a cause of disease in nonpregnant adults [1–3]. Invasive GBS (iGBS) disease in neonates and young infants can result from maternal colonisation and vertical transmission or environmental exposure after birth. It is classified by age at onset with early-onset GBS (EOGBS), occurring in the first 6 days of life, and late-onset disease (LOGBS), occurring between ages 7 and 89 days, and typically presents as sepsis, meningitis, or pneumonia. In 2020, an estimated 20 million pregnant women globally were colonised with GBS resulting in 231,000 (114,000 to 455,000) cases of EOGBS and a further 162,000 (70,000 to 394,000) LOGBS cases [1]. Together, these were estimated to have caused 58,000 to 91,000 infant deaths depending on the assumptions made about mortality in cases without access to healthcare. Furthermore, survivors of iGBS are at risk of long-term neurological sequelae with an estimated 37,000 (14,000 to 96,000) surviving infants developing moderate or severe neurodevelopmental impairment (NDI) [1,4]. Maternal colonisation with GBS is also an important cause of adverse pregnancy outcomes with an estimated 46,000 (20,000 to 111,000) GBS stillbirths and is potentially linked with 518,000 (36,000 to 1,142,000) excess preterm births.

Currently, the main strategies for preventing iGBS are based on intrapartum antibiotic prophylaxis (IAP). Many higher-income countries have reduced EOGBS incidence through IAP with eligible pregnant women identified either through risk factor–based screening or routine testing based on microbiological culture [5]. Despite this success, IAP has several limitations; notably, it is not effective against LOGBS- or GBS-associated stillbirths. In addition, the need for access to laboratory testing for microbiological screening–based strategies and the requirement to deliver antibiotics intravenously substantially limits the prospect of attaining high IAP coverage in many low-resource settings where the burden of iGBS is highest [5]. There are also

concerns that routine administration of antibiotics could contribute to antimicrobial resistance and might also have unintended impacts on the gut microbiota of newborns [6]. Hence, there is substantial interest in alternative approaches to prevention.

Maternal immunisation is a potential alternative strategy whereby vertical transfer of antibodies in utero from a woman vaccinated during pregnancy affords protection to the mother, unborn foetus, and newborn infant [7]. Maternal immunisation with Tetanus Toxoid has been successfully used to reduce the burden of neonatal tetanus since the 1970s, and, in the last decade, countries have been increasingly recommending routine vaccination of pregnant women against influenza and pertussis [8]. In 2015, development of a maternal vaccine against GBS was identified as a priority by the WHO Product Development for Vaccines Advisory Committee (PDVAC) [9], and 3 GBS maternal vaccine candidates have progressed to Phase II clinical trials [10]. In 2021, the licensure of an affordable GBS vaccine by 2026 was identified as a key milestone in the WHO global roadmap for Defeating Meningitis by 2030 [11].

There have been previous economic evaluations of maternal GBS vaccination in the United States [12–14], Europe [15–17], and sub-Saharan Africa [18–20]. However, none of these studies have estimated the value of GBS vaccination in all world regions. A global economic evaluation of GBS vaccination is important to drive investment into vaccine development by indicating the vaccine's potential value in different markets. It would also enable financing and pricing mechanisms to be put in place for equitable access to the vaccine once it is available. Such an evaluation is central to a Full Value of Vaccines Assessment (FVVA), which WHO has identified as key to catalysing vaccine development and subsequent equitable access [21,22]. To inform the WHO GBS vaccine FVVA [23], we conducted the first global economic evaluation of maternal GBS vaccination in 183 countries, drawing on recently updated global disease burden estimates for GBS [1].

## Methods

### Model overview

We developed a decision tree model (Fig 1) to assess the health impact and cost-effectiveness of maternal vaccination against GBS in an annual cohort of 140 million pregnant women and their babies for the year 2020 compared with current practice of no vaccination. The size of the cohort of pregnant women in each country was based on country-specific estimates of the number of births from the United Nations (UN) World Populations Prospects (UNWPP) [24]. Our analysis included the 183 countries out of 195 UN member states for which UNWPP birth data were available, which exclude countries with estimated populations below 90,000.

The health impact model structure was designed to reflect the natural history of pregnancy-related GBS infections and was aligned with the Bayesian disease modelling framework used in recently reported global estimates of GBS burden. The Bayesian disease model has been described in detail elsewhere; its structure is reflected in our methods below [1,25]. The model first stratifies pregnant women based on GBS colonisation status, and then by whether pregnancy results in a live birth or stillbirth. Live births are further subdivided into preterm and term births, with infants then at risk of developing either EOGBS or LOGBS; the risk of EOGBS among babies born to noncolonised mothers was assumed zero. iGBS disease (EOGBS or LOGBS) may then result in death or, among survivors of GBS sepsis or meningitis, either full recovery or long-term NDI.

The analysis used the lifetime of babies as the analytical time-horizon with health costs and quality-adjusted life years (QALYs) calculated over the lifetime of infant iGBS survivors using country-specific life expectancy at birth [24]. The model was used to compare scenarios with vaccination plus current practice against current practice without vaccination (i.e., assuming

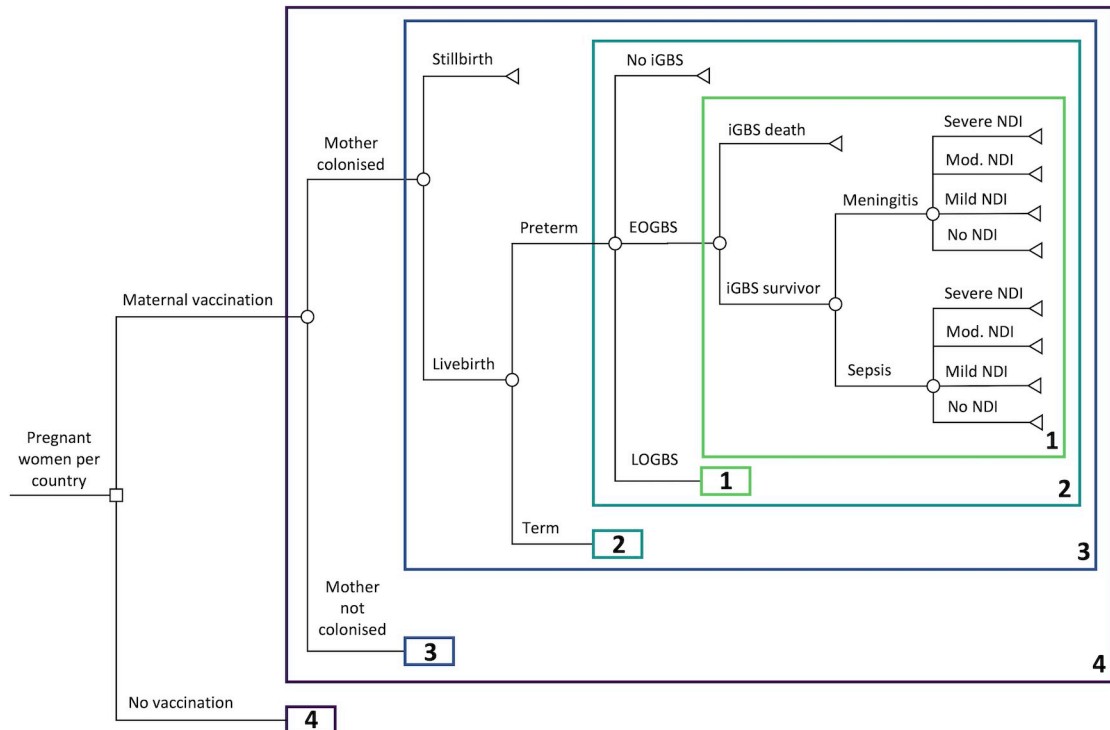

**Fig 1. Decision tree for GBS-related outcomes in children for an annual birth cohort in 183 countries comparing maternal vaccination against no vaccination (current standard of care).** Numbered boxes represent repeated model structure; however, the risks for some outcomes vary across repeated branches. EOGBS, early-onset GBS; GBS, Group B Streptococcus; iGBS, invasive GBS; LOGBS, late-onset GBS; NDI, neurodevelopmental impairment.

no change in each country's IAP policy following vaccine introduction). All analyses were performed using R version 4.0.2. This analysis is reported in accordance with the Consolidated Health Economic Evaluation Reporting Standards (CHEERS) checklist (S1 Appendix) and is informed by the burden estimates that were reported previously according to the GATHER statement.

## Disease risk

Model inputs are summarised in S1 Appendix A2. We parameterised the probabilities of different GBS-related outcomes in our model using posterior samples of key epidemiological parameters from the global burden estimates reported by Gonçalves and colleagues [1]. We used country-specific estimates of the prevalence of maternal GBS colonisation and of the risk of EOGBS in infants born to colonised mothers. The risks of LOGBS were then calculated using region-specific estimates of the fraction of iGBS cases that are EOGBS versus LOGBS. Regional classifications were based on the UN Sustainable Development Goal (SDG) region definitions [26].

Case fatality risks (CFRs) for EOGBS and LOGBS were also based on regional-level estimates from Gonçalves and colleagues. There are no data on CFRs for infants with EOGBS without access to care, so the authors considered 2 scenarios where they had either 90% CFR (following the approach of Seale and colleagues [3]) or the same CFR as other infants with EOGBS. In our analysis, we assumed in the base case that these infants had the same CFR as other infants with EOGBS, to be conservative about this highly uncertain parameter and because mothers of these children might also be less likely to receive maternal vaccines.

Among iGBS survivors, the proportion of sepsis and meningitis and the excess risk of mild, moderate, and severe NDI outcomes after meningitis were based on pooled global estimates, while NDI risks after sepsis were based on separate estimates for high-income and for low- and middle-income countries. The excess risk attributable to iGBS exposure was calculated assuming a counterfactual risk of mild or moderate and severe NDI among unexposed children from a large Danish cohort study [27]. We based the proportion of moderate and severe NDI that was severe on the same study. Following the approach used in the burden estimation, our base case analysis included only the excess risk of moderate or severe NDI, which is likely to be more consistent across settings, but include mild NDI as a sensitivity analysis [1,4].

To estimate country-specific GBS-associated stillbirth risk, national stillbirth estimates from the WHO Global Health Observatory [28] were combined with regional estimates of the proportion of stillbirths caused by GBS [1]. For the risk of GBS-associated prematurity, we used national data on the proportion of preterm births [29] together with the global odds ratio for the association between GBS maternal colonisation and preterm births [1]. Further details on these calculations are provided in S1 Appendix A2.3 and A2.4.

## Health outcomes

To calculate QALYs, we assumed country-specific life expectancy at birth for both normal births and survivors of iGBS and assigned zero QALYs for an iGBS death. For term births, we assumed no reduction in health-related quality of life (HRQoL), but for preterm births, we applied a utility decrement over the child's lifetime based on a systematic review and meta-analysis by Petrou and colleagues [30]. For the acute iGBS episode, we approximated QALY loss, assuming 29 days duration based on the average length of stay among studies in a recent systematic review of the acute costs of infant sepsis and meningitis [31], and applied health state utility decrements for hospitalisation with acute sepsis or meningitis from a US study in young children [32]. For survivors with long-term sequelae, we applied utility decrements from birth for mild, moderate, and severe NDI to each year of life and, conservatively, given previous studies provide evidence of post-acute mortality after bacterial meningitis [33,34], assumed no change in life expectancy. These utility values were based on a UK study, which assessed HRQoL in a cohort of children with NDI followed up at age 11 [35].

## Vaccination

Although clinical studies have demonstrated immunogenicity of candidate GBS vaccines, to date, there have been no Phase III efficacy trials [10]. We therefore based our assumptions about vaccine efficacy (VE) and other characteristics of a GBS vaccine on the WHO preferred product characteristics (PPCs) [36]. In our base case, we assumed a single-dose vaccine with 80% efficacy against both infant iGBS disease and GBS stillbirth across all GBS serotypes. We also assumed no effect on GBS-associated prematurity because (i) the WHO PPC does not specify that GBS vaccines must reduce colonisation, which is most likely pathway for preventing GBS-associated prematurity, and (ii) the association between GBS maternal colonisation and higher risk of prematurity may be confounded [37]. It is likely that delivery of GBS vaccines will need to be timed in either the late second trimester or third trimester and could be delivered through existing routine antenatal care (ANC) services. Hence, we assumed vaccine coverage based on the proportion of pregnant women in each country who attend at least 4 ANC visits (ANC4) [28].

We also considered a range of alternative scenarios (Table D in S1 Appendix): higher vaccine coverage based on the proportion of women attending at least 1 ANC visit (ANC1); a 2-dose regimen; lower and higher VE (60% and 90%); and a vaccine that is also effective

against GBS-associated prematurity. For the latter scenario, we estimated the proportion of preterm births that are potentially protected through vaccination by combining the distribution of preterm births by gestational age [38] with the timing of vaccine visits based on country-specific ANC data [39] (S1 Appendix A2.5).

## Costs

Our analysis was undertaken from a healthcare payer economic perspective, and all costs are reported in 2020 United States Dollars (USD). Where cost inputs were for different years, they were inflated using the World Bank Gross Domestic Product (GDP) deflator [40]. Costs reported in different currencies were then converted to 2020 USD using historical foreign exchange rates [41]. To estimate acute healthcare costs, we combined one GBS-specific cost estimate from a study in the United Kingdom (UK) [42], with the findings from a systematic review on the acute costs of infant sepsis and meningitis [43], and result of a recent study reporting the acute costs of neonatal bacterial sepsis and meningitis in Mozambique and South Africa [44]. We used linear regression to extrapolate country-specific cost estimates using total per capita healthcare expenditure as a predictor (S1 Appendix A2.6.) For long-term costs, no direct GBS-specific estimates exist in the literature. Annual costs among survivors with moderate and severe NDI were parameterised as a fixed proportion of between 4% and 28% of the acute cost estimate in each country, based on the range between a UK study of costs in children with NDI [35] and a US study of costs in adults with disabilities [45].

For the vaccine programme costs, we extrapolated results from a systematic review of maternal vaccination delivery costs using regression against GDP per capita (S1 Appendix A2.7) [31]. We used previously estimated vaccine prices by World Bank country income group, which were based on a combination of price benchmarking against other vaccines and cost of goods analysis: $50 for high-income countries; $15 for upper-middle-income countries; and $3.50 for lower-middle-income and low-income countries [46].

## Normative assumptions

A health intervention may be considered cost-effective if the cost per QALY gained falls below that country's cost-effectiveness threshold (CET). Here, we use 2 commonly cited thresholds: (i) country gross domestic product per capita [47]; and (ii) published thresholds based on empirical estimates of the health opportunity cost of healthcare spending (S1 Appendix A2.8) [48,49].

A second normative assumption is the QALY loss attributed to a stillbirth. In many settings, these are not assigned any health or disability weight, but it has been argued that they should be assigned a QALY loss close or the same as that of the death of a newborn [50]. Here, we consider 2 scenarios, one in which stillbirths are not assigned any QALY loss, and a second in which they are assigned the same QALY loss as the death of a newborn.

Following WHO guidelines, we discount costs at 3% and health effects at both 0% and 3% in alternative scenarios [51]. Table 1 summarises the normative scenarios used.

## Economic analysis

To assess the cost-effectiveness of GBS maternal vaccination compared to current practice, we follow a net monetary benefit (NMB) approach in which both the health and fiscal benefits of vaccination are expressed in monetary units [52]. To calculate the NMB, the incremental benefits in QALYs are multiplied by a country-specific CET (either empirical or 1 × GDP per capita) in USD and then the incremental costs are subtracted. An intervention may be considered

**Table 1. Parameter values used for least and most favourable normative assumptions.**

| Parameter | Least favourable assumptions | Most favourable assumptions |
|---|---|---|
| Discount rate | 3% for costs and benefits | 3% for costs; 0% for benefits |
| Inclusion of stillbirth QALYs | Not included | Included |
| CET | Based on empirical estimates* | 1 × GDP per capita |

*CETs were based either on estimates from Ochalek and colleagues or Woods and colleagues. See S1 Appendix A2.8 for more details.

CET, cost-effectiveness threshold; GDP, Gross Domestic Product; QALY, quality-adjusted life year.

cost-effective if the NMB is positive, since this is mathematically equivalent to the incremental cost-effectiveness ratio (ICER) being less than the CET.

An advantage of adopting an NMB framework is that our estimates for individual countries can be directly combined to estimate the aggregate value of vaccination both regionally and globally. We used multivariate linear regression to assess the influence of individual model parameters on the uncertainty of our estimates of global NMB. To account for combined parameter uncertainty, for each scenario, we ran 4,000 simulations per country sampling parameters from their corresponding probability distributions (Table D in S1 Appendix) and calculated the median and 95% uncertainty range (UR) based on 2.5 and 97.5 percentiles of the simulations. For costs, we used log-normal distributions; for health-state utility values, we used beta distributions; and for epidemiological parameters, we sampled from the posterior distributions from the Bayesian burden model. Due to lack of data, we assumed fixed values for the background risks of NDI after sepsis and meningitis not due to iGBS in line with the approach used in the global burden analysis. We also assumed fixed values for vaccine characteristics, since these are currently unknown, and we instead varied these in scenario analysis. Similarly, we used fixed vaccine prices, but as a sensitivity analysis estimated the threshold price at which a GBS vaccine would be cost-effective in each country. At the country level, we used the simulation results to calculate the probability maternal GBS vaccination is cost-effective (i.e., the proportion of simulations with NMB >0) under different scenarios.

## Results

We estimate that vaccinating 99.8 million pregnant women across 183 countries could cost $1.7 billion but could save around $300 million in acute healthcare costs and $85 million in long-term healthcare costs, although these estimates have wide uncertainty. Overall, the incremental cost of GBS vaccination is about $1.3 billion, with the biggest cost increase in Europe and Northern America (Table 2).

Globally, the vaccine programme could avert an estimated 127,000 (UR: 63,300 to 248,000) EOGBS cases and 87,300 (UR: 38,100 to 209,000) LOGBS cases, thus avoiding 31,100 (UR: 14,400 to 66,400) infant deaths and 17,900 (UR: 6,380 to 49,900) cases of moderate and severe NDI. Additionally, 23,000 (UR: 10,000 to 56,400) GBS stillbirths could be prevented, and, if a vaccine also proves effective against GBS-associated prematurity, 185,000 (UR: 13,500 to 407,000) preterm births might be avoided. The highest burden of iGBS cases and deaths, around two-fifths of the total, is averted in sub-Saharan Africa, which accounts for about one-fifth of the women vaccinated. In contrast, only about 1% of the deaths occur in Europe and Northern America despite a tenth of vaccinated women being in this region.

Overall, iGBS cases averted through vaccination resulted in a projected gain of 2.5 million (UR: 1.2 to 5.4 million) undiscounted QALYs, and a further 1.5 million (UR: 0.6 to 3.6 million) QALYs when avoided stillbirths are included. A vaccine that prevents GBS-associated

**Table 2. Annual global and regional impact of GBS maternal vaccination compared with no vaccination for the year 2020.**

| Description | Central and Southern Asia | Eastern and South-Eastern Asia | Europe and Northern America | Latin America and Caribbean | Northern Africa and Western Asia | Oceania | sub-Saharan Africa | Global^ |
|---|---|---|---|---|---|---|---|---|
| Number of women vaccinated (millions) | 22.8 | 25.5 | 11.7 | 9.51 | 7.8 | 0.546 | 21.9 | 99.8 |
| Vaccine programme costs (discounted; $ millions) | 124 (117, 136) | 470 (452, 495) | 648 (621, 687) | 173 (169, 178) | 127 (124, 131) | 24.4 (22.9, 26.6) | 107 (104, 112) | 1,680 (1,640, 1,720) |
| Acute healthcare costs (discounted; $ millions) | −7.93 (−17.4, −3.74) | −54.6 (−119, −24.9) | −155 (−352, −60.9) | −21.4 (−47.5, −10.7) | −27.8 (−56.9, −13.5) | −3.89 (−8.87, −1.57) | −14.3 (−30.5, −6.67) | −298 (−534, −155) |
| Long-term healthcare costs (discounted; $ millions) | −2.95 (−11.3, −0.581) | −19.8 (−79.2, −3.78) | −33.2 (−117, −7.02) | −8.13 (−31.1, −1.63) | −10 (−36.6, −2.08) | −0.871 (−2.97, −0.185) | −4.85 (−18.6, −1.01) | −86.5 (−252, −20.6) |
| Total incremental costs (discounted; $ millions) | 113 (96.6, 127) | 394 (286, 446) | 456 (200, 581) | 143 (99.5, 160) | 88.9 (39.5, 110) | 19.6 (13, 23.2) | 87.9 (60.6, 99.7) | 1,290 (948, 1,490) |
| EOGBS cases (thousands) | −22.8 (−43.6, −11.6) | −30.6 (−62.4, −14.5) | −3.37 (−5.94, −1.57) | −8.91 (−16.8, −4.72) | −17.9 (−36, −8.6) | −0.421 (−0.86, −0.207) | −42.3 (−86.4, −20.1) | −127 (−248, −63.3) |
| LOGBS cases (thousands) | −11.3 (−33.1, −2.94) | −15.2 (−45.7, −3.86) | −1.99 (−4.2, −0.84) | −5.9 (−20.4, −1.94) | −12.9 (−33, −5.24) | −0.275 (−1.39, −0.0888) | −36.4 (−101, −14) | −87.3 (−209, −38.1) |
| Moderate & severe NDI cases (thousands) | −2.91 (−8.52, −0.933) | −3.86 (−11.8, −1.21) | −0.257 (−0.572, −0.0975) | −1.31 (−3.99, −0.42) | −2.63 (−7.58, −0.9) | −0.0541 (−0.223, −0.0182) | −6.66 (−19.4, −2.2) | −17.9 (−49.9, −6.38) |
| GBS deaths (thousands) | −4.22 (−9.6, −1.78) | −5.58 (−13.4, −2.32) | −0.335 (−0.668, −0.148) | −1.95 (−4.7, −0.804) | −4.76 (−10.8, −2.06) | −0.0836 (−0.336, −0.0242) | −13.6 (−32.2, −5.65) | −31.1 (−66.4, −14.4) |
| GBS stillbirths (thousands) | −6.59 (−23, −1.63) | −3.13 (−10.9, −0.794) | −0.586 (−1.48, −0.206) | −1.34 (−8.35, −0.222) | −1.33 (−3.12, −0.591) | −0.0646 (−0.336, −0.0174) | −9.31 (−18.5, −4.12) | −23 (−56.4, −10) |
| GBS associated preterm births* (thousands) | −34.7 (−78.5, −2.33) | −28.7 (−66.1, −2.05) | −33.2 (−72, −2.4) | −18.5 (−40.9, −1.37) | −23.8 (−52.4, −1.81) | −1.45 (−3.27, −0.103) | −44.4 (−97.4, −3.23) | −185 (−407, −13.5) |
| QALYs from averted GBS disease (discounted; thousands) | 150 (66.3, 339) | 203 (86.9, 485) | 12.5 (5.64, 24.8) | 70.2 (30.7, 168) | 164 (73, 371) | 2.96 (1.01, 10.6) | 431 (186, 1,010) | 1,060 (486, 2,270) |
| QALYs from averted stillbirths (discounted; thousands) | 180 (43.7, 627) | 87.2 (21.7, 301) | 16.5 (5.77, 41.3) | 37.3 (6.18, 233) | 36.9 (16.3, 85.9) | 1.75 (0.474, 9.06) | 243 (108, 486) | 622 (271, 1,550) |
| QALYs from averted preterm births* (discounted; thousands) | 63.6 (5.01, 162) | 53.8 (4.24, 143) | 62.4 (4.99, 160) | 34.6 (2.77, 88.8) | 44.2 (3.58, 113) | 2.77 (0.213, 7.23) | 77.4 (6.26, 198) | 338 (27.6, 857) |
| QALYs from averted GBS disease (undiscounted; thousands) | 358 (158, 807) | 513 (220, 1,220) | 32.7 (14.6, 64.6) | 177 (77, 422) | 408 (181, 923) | 6.94 (2.46, 24.5) | 947 (407, 2,230) | 2,490 (1,160, 5,370) |
| QALYs from averted stillbirths (undiscounted; thousands) | 429 (104, 1,490) | 218 (54.4, 752) | 42.7 (15, 107) | 94.1 (15.6, 586) | 91.4 (40.4, 213) | 4.19 (1.17, 20.9) | 532 (237, 1,060) | 1,460 (630, 3,670) |
| QALYs from averted preterm births* (undiscounted; thousands) | 152 (12, 387) | 135 (10.7, 359) | 163 (13, 417) | 87.8 (7.03, 225) | 110 (8.91, 282) | 7.3 (0.56, 19) | 171 (13.8, 436) | 825 (67.4, 2,090) |

All values are reported to 3 significant figures. Values in brackets are 95% URs.

EOGBS, early-onset GBS; GBS, Group B Streptococcus; LOGBS, late-onset GBS; NDI, neurodevelopmental impairment; QALY, quality-adjusted life year; UR, uncertainty range; VE, vaccine efficacy.

*In scenario analysis where vaccine is assumed to have 80% VE against GBS-associated prematurity.

^Global median values do not exactly equal the sum of the regional median values.

prematurity could add another 0.8 million (UR: 0.1 to 2.1 million) QALYs. The relative contribution of preventing iGBS, stillbirths, and prematurity to the overall QALY gain varies by region. For example, in sub-Saharan Africa and Northern Africa and Western Asia, preventing iGBS contributes the majority of the QALY gain, but in Europe and Northern America and Central and Southern Asia, avoided stillbirths make a larger contribution. In Europe and Northern America, preventing preterm births might result in larger QALY gains than iGBS cases and stillbirths combined.

Using our base case assumptions about the vaccine characteristics, the estimated global NMB of vaccination ranged from $1.1 billion (UR: $−0.2 to 3.9 billion) under least favourable normative assumptions to $17 billion (UR: $9.1 to 31 billion) under most favourable normative assumptions (Table 1 and Fig 2A). Including stillbirth QALYs increases the NMB by between $1.4 billion and $7.1 billion depending on the other normative assumptions made. Although the point estimate was positive under all assumptions, for the least favourable assumptions, the UR includes zero.

Under the most favourable normative assumptions, vaccination had a positive NMB in all regions (Fig 2B). However, for least favourable assumptions, the NMB was negative for Central and Southern Asia, Europe and Northern America, and Oceania. Nevertheless, if stillbirth QALYs were included, the NMB for these regions were again positive (Fig A in S1 Appendix).

For the most favourable normative assumptions, vaccination is likely cost-effective (mean NMB >0) in almost all countries (Fig 3), but for least favourable assumptions, this was reduced to around 60% (112/183) of countries. Notably, vaccination was less likely to be cost-effective among countries with lower GDP per capita within the sub-Saharan Africa, Central and Southern Asia, and Europe and Northern America regions.

Fig G in S1 Appendix shows the model input parameters that had the greatest influence on the uncertainty of the NMB estimates under least favourable normative assumptions. Globally, NMB was most sensitive to variation in the risk of EOGBS (particularly in countries with higher numbers of births and higher GDPPC, e.g., China and USA), the proportion of iGBS that is late-onset, CFR, and the risk of developing moderate/severe NDI. Fig 4 shows how the global NMB of vaccination varies under different scenarios. Inclusion of mild NDI, assuming GBS births without skilled birth attendants have 90% case fatality or increasing VE from 80% to 90% slightly increase the global NMB of vaccination, while assuming zero long-term costs for NDI, slightly decreases the NMB. However, none of these assumptions have a dramatic effect. If VE is decreased to 60%, global NMB remains positive under least favourable assumptions, but the number of countries for which vaccination is no longer cost-effective increases to 91 (Fig D in S1 Appendix). A vaccine that requires 2 doses to achieve 80% efficacy would have a negative global NMB, and vaccination would not be cost-effective in 110 countries. However, a vaccine with protection against preterm birth substantially increases the global NMB and is especially influential in the Europe and Northern America region (see Fig B in S1 Appendix).

The distribution of vaccine threshold prices among countries within each World Bank income group are shown in Fig 5 (results by SDG region are shown in Fig E in S1 Appendix, and for other vaccine scenarios in Fig F in S1 Appendix). The threshold price is usually positive (i.e., there is some price at which purchasing the vaccine would be cost-effective) and generally higher in high-income and upper-middle-income countries. However, under least favourable normative assumptions, threshold price is negative in 8 countries, indicating that even with a free vaccine the delivery costs outweigh the health benefits in this analysis.

## Discussion

A high-coverage global maternal immunisation programme against GBS could avert hundreds of thousands of GBS cases, alongside tens of thousands of deaths, stillbirths, and cases of long-

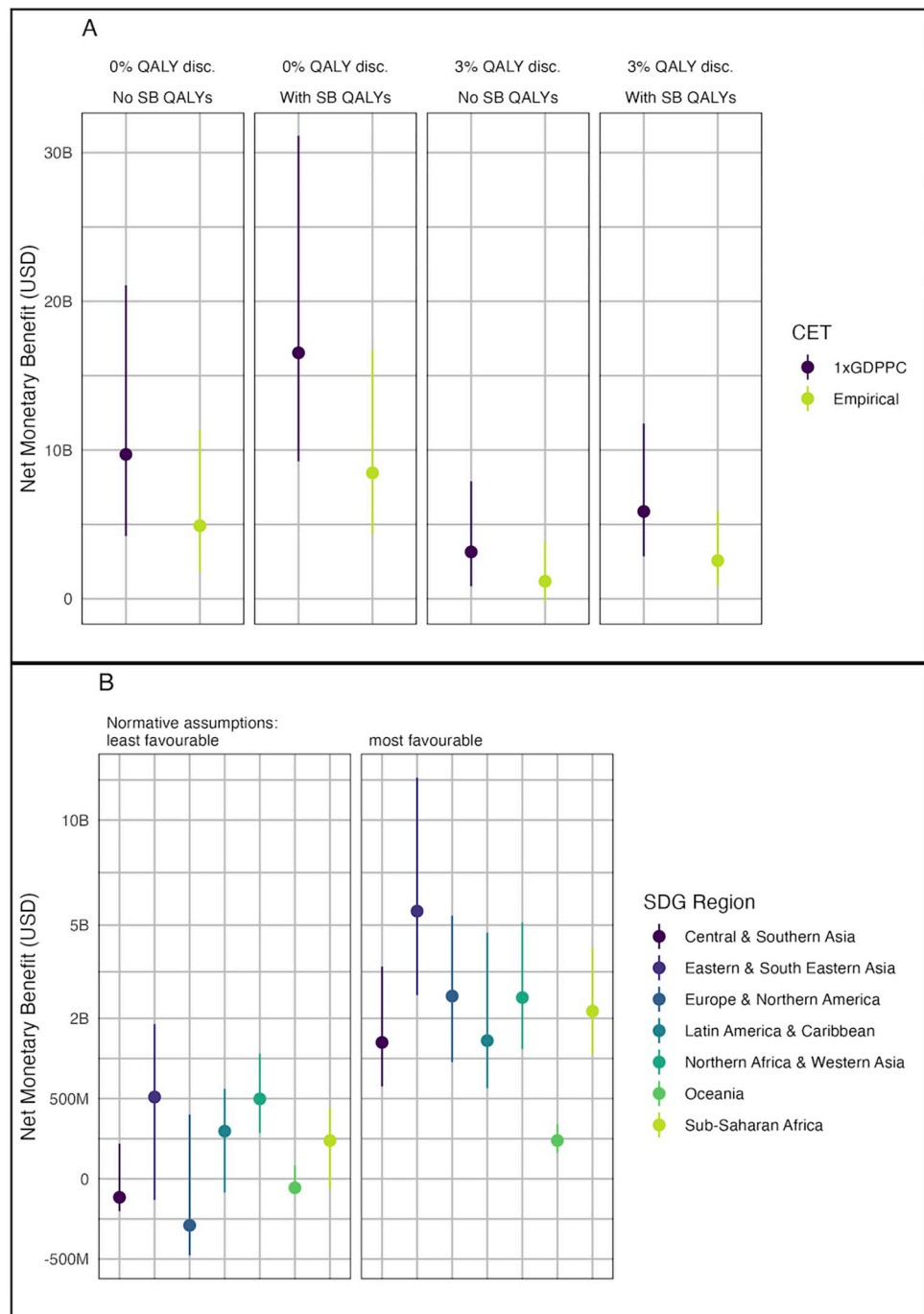

**Fig 2. NMB of GBS maternal vaccination (A) globally under different normative assumptions (see Table 1) and (B) by region for the most and least favourable normative assumptions.** Least favourable normative assumptions were the use of an empirical CET, 3% discounting of QALYs, and exclusion of stillbirth QALYs. Most favourable assumptions were the use of 1 × GDP per capita CETs, 0% discounting of QALYs, and inclusion of stillbirth QALYs. M, Millions; B, Billions; CET, cost-effectiveness threshold; GDP, Gross Domestic Product; GDPPC, GDP per capita; NMB, net monetary benefit; QALY, quality-adjusted life year; SB, stillbirth; SDG, Sustainable Development Goal.

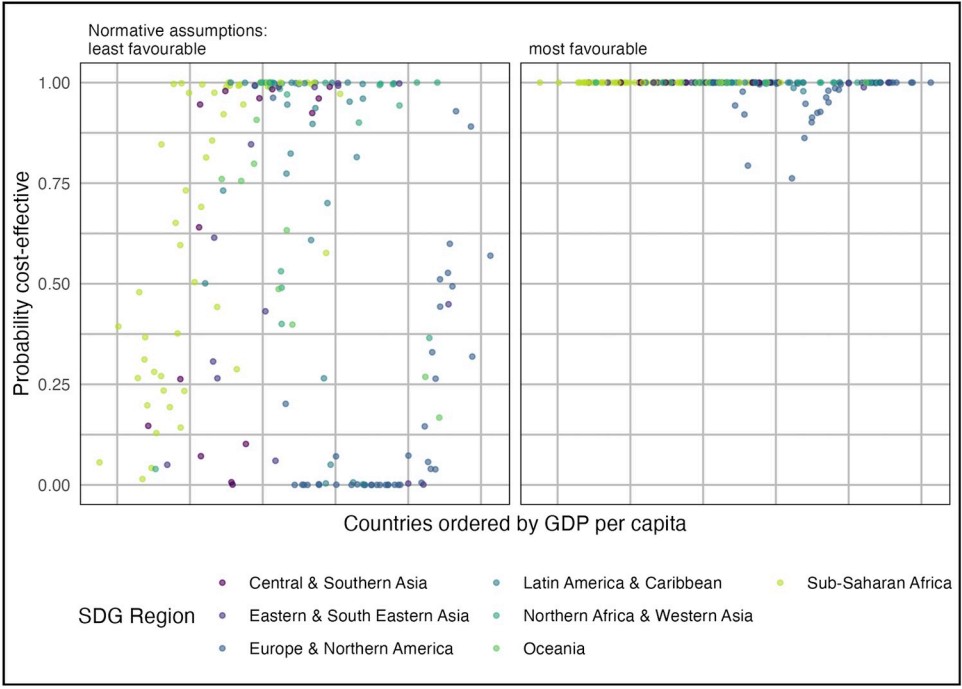

**Fig 3. Probability that GBS maternal vaccination is cost-effective in each country under most favourable and least favourable normative assumptions (see Table 1).** Least favourable normative assumptions were the use of an empirical CET, 3% discounting of QALYs, and exclusion of stillbirth QALYs. Most favourable assumptions were the use of 1 × GDP per capita CETs, 0% discounting of QALYs, and inclusion of stillbirth QALYs. CET, cost-effectiveness threshold; GBS, Group B Streptococcus; GDP, Gross Domestic Product; GDPPC, GDP per capita; QALY, quality-adjusted life year; SB, stillbirth; SDG, Sustainable Development Goal.

term disability. We estimate that such a programme may have a net cost of around $1.3 billion, with most costs occurring in Europe and Northern America. Nevertheless, it would be cost-effective in most countries under favourable assumptions, particularly if it can reduce preterm births.

Even under less favourable assumptions, a single-dose GBS vaccine could still be cost-effective due to additional factors we did not explore. In some high-income countries, GBS vaccination plus current practice may be less cost-effective compared to current practice alone because of lower GBS incidence in babies due to IAP. However, GBS vaccination might allow high-income countries to achieve additional cost savings by revising IAP algorithms for vaccinated mothers. In low- and lower-middle-income countries, iGBS incidence may be higher, but so is the health opportunity cost of healthcare spending due to budget constraints leading to lower thresholds at which interventions may be considered cost-effective.

More competitive pricing may enable vaccination to be cost-effective, even under least favourable assumptions. Competitive and finely tiered vaccine prices could also be beneficial for manufacturers, with financial analyses suggesting that high global demand is needed to ensure the development costs of a GBS vaccine can be recouped [46]. Our economic evaluation can inform both manufacturers and donors investigating the financial viability of investing in GBS vaccine development, as well as countries identifying the price they should be willing to pay for such a vaccine.

To our knowledge, our study is the first to estimate the value of maternal GBS vaccination across all regions and country income groups. Previous analyses have estimated cost-effectiveness in the US [12–14], the Netherlands [17], UK [15,16], South Africa [19], The Gambia [18],

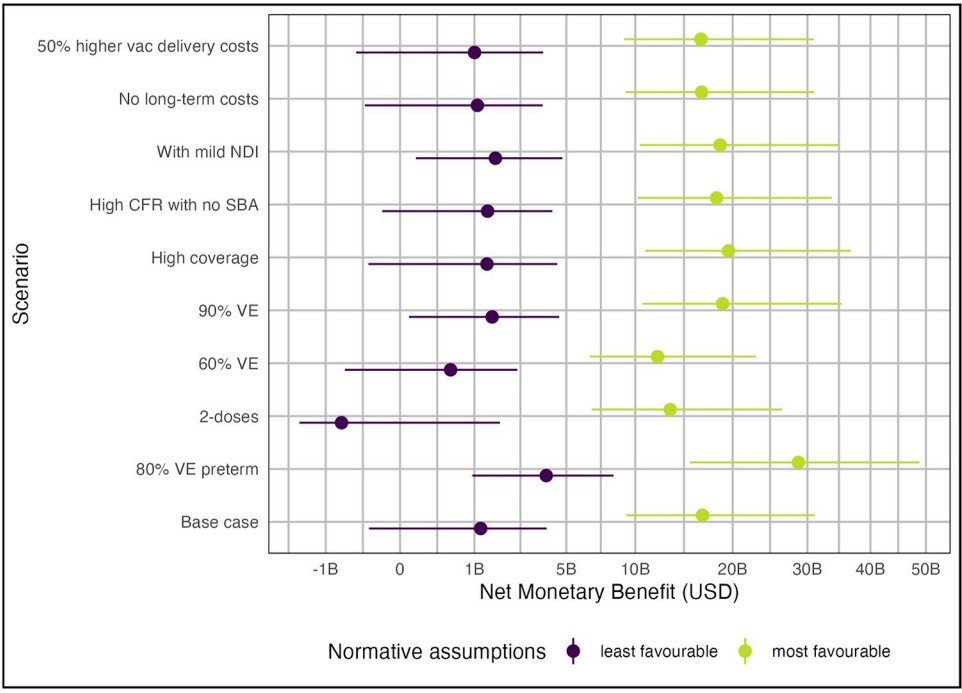

**Fig 4. Annual global NMB of GBS maternal vaccination under most favourable and least favourable normative assumptions (see Table 1) for different vaccination scenarios.** Points show median estimates and lines show 95% URs. Least favourable normative assumptions were the use of an empirical CET, 3% discounting of QALYs, and exclusion of stillbirth QALYs. Most favourable assumptions were the use of 1 × GDP per capita CETs, 0% discounting of QALYs, and inclusion of stillbirth QALYs. B, Billions; CET, cost-effectiveness threshold; CFR, case fatality risk; GDP, Gross Domestic Product; NDI, neurodevelopmental impairment; QALY, quality-adjusted life year; SBA, skilled birth attendant; VE, vaccine effectiveness.

and 37 Gavi countries in Africa [20]. These prior estimates suggested cost-effectiveness of vaccination ranged from $320 to 573 per DALY averted in Gavi-eligible countries [20], to $3,550 per DALY averted in South Africa [19], to over $50,000 per QALY in the US [12,13], which is broadly consistent with our results. Like our analysis, Kim and colleagues also found that the ability to avert GBS-associated prematurity greatly improved vaccine cost-effectiveness [19].

As far as we know, this was the first cost-effectiveness study to use new global estimates of the health burden due to GBS including infant morbidity and mortality, long-term NDI, stillbirth, and GBS-associated prematurity. This burden study propagated parametric uncertainty comprehensively by using a Bayesian framework to synthesise existing data sources. Posterior distributions from the study then informed a probabilistic sensitivity analysis for our cost-effectiveness model. Similarly, for cost data, parameters with multiple sources of data from previous systematic reviews were synthesised using regression models. Conversely, the main limitations of our analysis reflected parameters with limited data such as those governing health-related quality of life and long-term costs from disability, where estimates were based on only 1 to 2 relevant studies. Our analysis also excluded the potential impact of vaccination on maternal morbidity and the costs of GBS-related disability beyond the health-sector. However, both these factors would likely reinforce our main findings on cost-effectiveness.

A further set of uncertainties govern GBS vaccine characteristics such as efficacy, number of doses needed, and impact on GBS-associated prematurity. Since there is currently no licensed vaccine, these parameters were informed by the WHO PPC, which is based on expert assumptions. We therefore used scenario sensitivity analyses to identify which of these

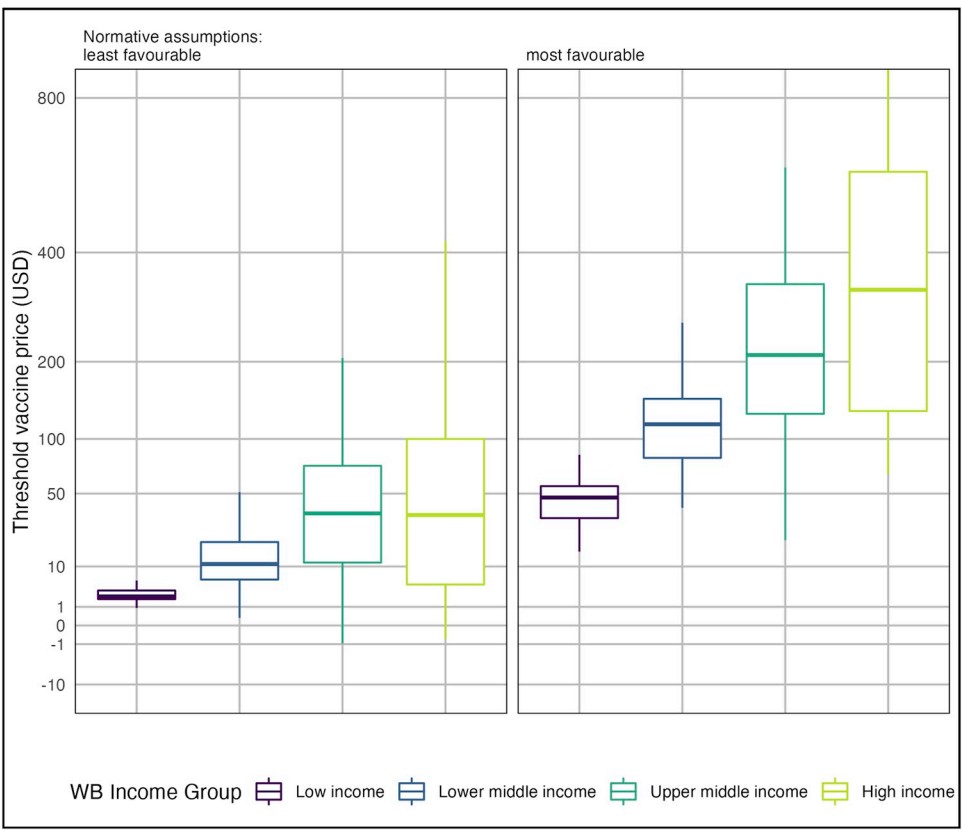

**Fig 5. Distribution of GBS vaccine threshold prices among countries within each World Bank income group under most and least favourable normative assumptions (see Table 1).** Threshold vaccine prices above $800 per dose are not shown. Least favourable normative assumptions were the use of an empirical CET, 3% discounting of QALYs, and exclusion of stillbirth QALYs. Most favourable assumptions were the use of 1 × GDP per capita CETs, 0% discounting of QALYs, and inclusion of stillbirth QALYs. CET, cost-effectiveness threshold; GBS, Group B Streptococcus; GDP, Gross Domestic Product; QALY, quality-adjusted life year.

characteristics are the most important drivers of vaccine value. Further data from carefully designed vaccine trials and other field studies are needed to inform these data gaps. Vaccine value is also driven by normative health economic assumptions around discounting, CETs, and the value of preventing stillbirths, which reflect uncertainty about the values of society rather than about empirical data. As GBS vaccines progress towards licensure, and new data on vaccine characteristics and GBS epidemiology become available, updated analyses tailored to individual country contexts can help ensure that vaccines are acquired at prices that are cost-effective.

Overall, our results suggest high coverage of a competitively priced maternal GBS vaccine has the potential to save tens of thousands of lives globally and is likely to be a cost-effective investment, particularly if the vaccine can reduce GBS-associated prematurity.

## Supporting information

**S1 Appendix. Supplementary appendix. Table A. Updated Consolidated Health Economic Evaluation Reporting Standards (CHEERS) checklist from [53]. Table B. Model parameter values used in base case analysis**. (**a**) For parameters estimated in our previously published burden model, we used samples of the posterior distribution of the parameter estimates, and

these samples along with other model input data are provided online. Values presented in this table correspond to the mean of the posterior distributions. (**b**) Regions for these parameters are based on a country's World Bank income classification. (**c**) See section A2.7. (**d**) See section A2.6. (**e**) For these parameters, countries are assigned to the "developed" region according to the World Bank development status; for countries not classified as "developed," the region is based on the UN geographical region. (**f**) Regions for these parameters are based on a country's World Bank development status. (**g**) In the base case, we conservatively assume no excess risk of mild NDI following GBS by setting the value for these parameters to the match the baseline risk of mild NDI, i.e., we assume $Risk_{mild-NDI-sep} = Risk_{mild-NDI-men} = Risk_{mild-NDI-baseline}$. EOGBS, early-onset GBS; GBS, Group B Streptococcus; HIC, high-income country; iGBS, invasive GBS; LIC, low-income country; LMIC, lower middle-income country; LOGBS, late-onset GBS; NDI, neurodevelopmental impairment; OR, odds ratio; UMIC, upper middle-income country. **Table C. Country-specific model inputs values**. (**a**) Estimated using regression model (see section A2.6 for further details). (**b**) Estimated using regression model (see section A2.7 for further details). (**c**) Values in italics were based on the average across other high-income countries. (**d**) See section A2.5 for further details on how these values were estimated. (**e**) Based on estimates World Bank where available, otherwise based on estimated from the IMF. (**f**) Based on values from Ochalek and colleagues and Woods and colleagues—see section A2.8 for how these values were calculated; values in italics were imputed using regression against GDP per capita. ANC, antenatal care; CET, cost-effectiveness threshold; EOGBS, early-onset Group B Streptococcus disease; GDP, Gross Domestic Product; USD, United States Dollars. **Table D. Changes to base case parameter assumptions used in scenario analysis. Table E. Distribution of preterm births by gestational age.** The proportion of preterm births occurring by week of gestational age is based on the global values reported by Blencowe and colleagues [54] **Table F. Country-specific estimates of the proportion of pregnant women vaccinated by gestation age in weeks.** These estimates were provided by the authors of [39] and were calculated using input data on antenatal coverage that was available at the time of their analysis. **Table G. Cost data used to extrapolate country-specific estimates of the acute healthcare costs of the iGBS episode. Table H. Cost data from [31] used to extrapolate used to extrapolate country-specific estimates of vaccine delivery costs per dose. Table I. Annual global and regional incremental impact of GBS vaccination compared with no vaccination for 2020 under high-coverage scenario. Fig A. Regional NMB of GBS vaccination under different normative assumptions about the discount rate for QALYs, the cost-effectiveness threshold, and whether the value of QALYs for averted stillbirths are included.** M, Millions; B, Billions; CET, cost-effectiveness threshold; GDP, Gross Domestic Product; GDPPC, GDP per capita; NMB, net monetary benefit; QALY, quality-adjusted life year; SB, stillbirth; SDG, Sustainable Development Goal. **Fig B. Regional NMB of different GBS maternal vaccination scenarios under the most and least favourable normative assumptions.** Least favourable normative assumptions were the use of an empirical CET, 3% discounting of QALYs, and exclusion of stillbirth QALYs. Most favourable assumptions were the use of $1 \times$ GDP per capita CETs, 0% discounting of QALYs, and inclusion of stillbirth QALYs. M, Millions; B, Billions; CET, cost-effectiveness threshold; GDP, Gross Domestic Product; GDPPC, GDP per capita; NMB, net monetary benefit; QALY, quality-adjusted life year; SB, stillbirth; SDG, Sustainable Development Goal. **Fig C. Probability that GBS maternal vaccination is cost-effective in each country for base case vaccination scenario and different normative assumptions about the discount rate for QALYs, the cost-effectiveness threshold, and whether the value of QALYs for averted stillbirths are included.** M, Millions; B, Billions; CET, cost-effectiveness threshold; GBS, Group B Streptococcus; GDP, Gross Domestic Product; GDPPC, GDP per capita; QALY, quality-adjusted life year; SB, stillbirth;

SDG, Sustainable Development Goal. **Fig D. Probability that GBS maternal vaccination is cost-effective in each country for different vaccination scenarios under most favourable and least favourable normative assumptions.** Least favourable normative assumptions were the use of an empirical CET, 3% discounting of QALYs, and exclusion of stillbirth QALYs. Most favourable assumptions were the use of 1 × GDP per capita CETs, 0% discounting of QALYs, and inclusion of stillbirth QALYs. CET, cost-effectiveness threshold; GBS, Group B Streptococcus; GDP, Gross Domestic Product; GDPPC, GDP per capita; QALY, quality-adjusted life year; SB, stillbirth; SDG, Sustainable Development Goal. **Fig E. Distribution of GBS vaccine threshold prices among countries within each SDG region under most and least favourable normative assumptions** Threshold vaccine prices above $800 per dose are not shown. **Fig F. Distribution of GBS vaccine threshold prices among countries within each World Bank region under most and least favourable normative assumptions for different vaccination scenarios.** Threshold vaccine prices above $800 per dose are not shown. Least favourable normative assumptions were the use of an empirical CET, 3% discounting of QALYs, and exclusion of stillbirth QALYs. Most favourable assumptions were the use of 1 × GDP per capita CETs, 0% discounting of QALYs, and inclusion of stillbirth QALYs. B, Billions; CET, cost-effectiveness threshold; CFR, case fatality risk; GDP, Gross Domestic Product; NDI, neurodevelopmental impairment; SBA, skilled birth attendant; QALY, quality-adjusted life year; VE, vaccine effectiveness. **Fig G. Tornado diagram showing impact of varying individual model parameters on the estimated global NMB using the least favourable normative assumptions.** Only the top 100 most influential parameters are shown. The impact of varying each parameter value between the 2.5% and 97.5% quantiles (shown by the different colour bars) was estimated using multivariables linear regression. Regional parameters are shown by the region name in parentheses, country-level parameters by a suffix with the country iso3 code, and all other parameters are global. For the regression, the risk of maternal colonisation and risk of EOGBS given colonisation were combined into a single risk of EOGBS parameter for each country to preserve the correlation between these 2 jointly sampled parameters. Least favourable normative assumptions were the use of an empirical CET, 3% discounting of QALYs, and exclusion of stillbirth QALYs. CET, cost-effectiveness threshold; EOGBS, early-onset GBS disease; LOGBS, late-onset GBS disease; NDI, neurodevelopmental impairment.
(DOCX)

## Acknowledgments

We would like to thank the authors of the GBS burden paper for sharing data on the posterior estimates of parameters in the burden model. We thank the GBS Full Value of Vaccine Assessment project Scientific Advisory Group for helpful discussion. We also thank Clint Pecenka and Ranju Baral for sharing estimates of antenatal care coverage by gestational age.

The views expressed in this article are those of the authors and do not necessarily represent the decisions, official policy, or opinions of WHO.

## Author Contributions

**Conceptualization:** Simon R. Procter, Artemis Koukounari, Joy E. Lawn, Mark Jit.

**Data curation:** Simon R. Procter, Bronner P. Gonçalves.

**Formal analysis:** Simon R. Procter.

**Funding acquisition:** Joy E. Lawn, Mark Jit.

**Investigation:** Simon R. Procter.

**Methodology:** Simon R. Procter, Bronner P. Gonçalves, Caroline Trotter, Mark Jit.

**Software:** Simon R. Procter.

**Supervision:** Joy E. Lawn, Mark Jit.

**Validation:** Simon R. Procter, Bronner P. Gonçalves.

**Visualization:** Simon R. Procter.

**Writing – original draft:** Simon R. Procter, Mark Jit.

**Writing – review & editing:** Simon R. Procter, Bronner P. Gonçalves, Proma Paul, Jaya Chandna, Farah Seedat, Artemis Koukounari, Raymond Hutubessy, Caroline Trotter, Joy E. Lawn, Mark Jit.

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
