## [Editor Report · Decision Letter 0]

8 Jul 2022

Dear Dr Procter, 

Thank you for submitting your manuscript entitled "Maternal immunisation against Group B Streptococcus: a global analysis of health impact and cost-effectiveness" for consideration by PLOS Medicine.

Your manuscript has now been evaluated by the PLOS Medicine editorial staff as well as by an academic editor with relevant expertise and I am writing to let you know that we would like to send your submission out for external peer review.

Please re-submit your manuscript within two working days, i.e. by Jul 12 2022 11:59PM.

Kind regards,

Philippa

Dr Philippa Dodd, MBBS MRCP PhD

Senior Editor

PLOS Medicine

---

## [Decision Letter · Decision Letter 1]

13 Oct 2022

Dear Dr. Procter,

Thank you very much for submitting your manuscript "Maternal immunisation against Group B Streptococcus: a global analysis of health impact and cost-effectiveness" (PMEDICINE-D-22-02313R1) for consideration at PLOS Medicine. 

[LINK]

In light of these reviews, I am afraid that we will not be able to accept the manuscript for publication in the journal in its current form, but we would like to consider a revised version that addresses the reviewers' and editors' comments. Obviously we cannot make any decision about publication until we have seen the revised manuscript and your response, and we plan to seek re-review by one or more of the reviewers. 

We expect to receive your revised manuscript by Nov 03 2022 11:59PM. Please email us (plosmedicine@plos.org) if you have any questions or concerns.

We look forward to receiving your revised manuscript. 

Sincerely,

Pippa

Philippa Dodd, MBBS MRCP PhD

PLOS Medicine

pdodd@plos.org

plosmedicine.org

Apologies for the delay in returning the manuscript to you. It took a long time to get reviewed, largely as result of multiple COIs. I addition, we are also unexpectedly extremely short staffed at the moment. The manuscript was reviewed by a subject reviewer, a methodological reviewer and an academic editor with relevant expertise. Their comments are included below.

GENERAL

Please address all editor and reviewer comments below.

Thank you for reporting your study according to the CHEERS 2022 statement. Given the nature of your study, it might be worth considering also the GATHER statement, further guidance can be found here: http://gather-statement.org/

ABSTRACT

* Please structure your abstract using the PLOS Medicine headings (Background, Methods and Findings, Conclusions). 

* Please combine the Methods and Findings sections into one section, “Methods and findings”.

Abstract Methods and Findings: 

* Please ensure that all numbers presented in the abstract are present and identical to numbers presented in the main manuscript text. 

* Please expand the description of the study design which is rather brief – size of female cohort? any important dependent variables that are adjusted for/considered? 

Abstract Conclusion

Please remove the heading “interpretation” and instead include a conclusions section considering the following points:

* Please address the study implications without overreaching what can be concluded from the data; the phrase "In this study, we observed ..." may be useful. 

* Please interpret the study based on the results presented in the abstract, emphasizing what is new without overstating your conclusions. 

* Please avoid vague statements such as "these results have major implications for policy/clinical care". Mention only specific implications substantiated by the results. 

* Please avoid any assertions of primacy ("We report for the first time....")

AUTHOR SUMMARY

METHODS and RESULTS

Ordinarily we would ask that main results are quantified with 95% CIs and p-values (both in the abstract and in the main manuscript). I understand the use of uncertainty intervals/ranges in a study of this kind. Where possible and appropriate, please also include p-values, if not please give a brief justification. If/where p-values are presented please also provide the statistical test used to determine them. 

Line 107 “We developed a decision-tree model (Fig. 1)” please revise to “Figure. 1” Please check and amend throughout the manuscript.

Line 109 “The size of the cohort of women in each country was calculated by…”. Its not immediately obvious what the cohort size was, it takes a bit of looking and reading of different parts to appreciate this. Suggest revising for improved clarity/reader accessibility

Line 267 please remove the funding statement from this section and include in the manuscript submission form

FIGURES

These are very nicely presented and well described. Throughout including supplementary files, please consider making them accessible to those with colour blindness (i.e. avoiding the use of red/green)

TABLES

Where appropriate please also include p-values (as above) 

DISCUSSION

Please ensure the Discussion is presented as follows: a short, clear summary of the article's findings; what the study adds to existing research and where and why the results may differ from previous research; strengths and limitations of the study; implications and next steps for research, clinical practice, and/or public policy; one-paragraph conclusion.

Line 435 – please remove the declaration of interests statement and include only in the submission form 

REFERENCES

Please select the PLOS Medicine reference style in your citation manager. For in-text reference call outs, citations should be in square brackets, and preceding punctuation, note the absence of spaces within the square brackets, “…symptomatic [2,8].” 

In the bibliography please ensure no more than 6 authors are listed before et al where more than n=6 authors contribute. Journal name abbreviations should be those found in the National Center for Biotechnology Information (NCBI) databases. 

Comments from the academic editor:

I have now read the reviewers' comments and the submitted paper and agree with the suggestion of (to me) major changes. Reviewer 3 in particular gives comments that would require some additional analysis which is entirely doable.

It is an interesting paper focussing on an important public health infection - although given that no vaccine is currently available this is a bit hypothetical but could push the vaccine development agenda.

I look forward to seeing the revised manuscript.

Comments from the reviewers:

Reviewer #1: This is a well-conducted global cost-effectivenes analysis on the impact of maternal immunisation against Group B Streptococcus. The study design, datasets, statistical methods and analyses, and presentation (tables and figures) and interpretation of the results are mostly adequate and of a very good standard. However, there are still a few minor points needing attention.

1) One of the key findings is "Estimated global NMB ranged from $1.1 billion ($-0.2 - 3.8 billion) to $17 billion ($9.1 - 31 billion)", which came from the results in Figure 4. The readers could try their best to find this conclusion from Figure 4, however as this is such an important result, it would be good for authors make it absolutely clear and highlighted in the main text on exactly where this lower limit (1.1 billion) and higer limit (17 billion) came from and under what assumption and senario. Also need to mention, the 95% CI for the lower bound estimate includes 0, which strictly speaking is not statistically significant - meaning under least favourable assumption/senario the proposed vaccine may not be cost-effective, which needs to be discussed in balance with the positive results under the favourable situation.

2) Figure 4. Under the least favourable assumption, when looking at the 95% CI of many scenarios, the NMBs are not statistically significant (being positive) such as for 60% VE, 2 dose, high coverage and base case, which means the vaccine may not be cost-effective under these circumstances. I can see the authors have discussed the uncertainties in the discussion, however it would be good to go a bit further on how to make an effective implementation of the proposed vaccine program to avoid these pitfalls given that the sensitivity/scenario analyses have already usefully identified those problems.

Reviewer #2: Thank you for your revised manuscript. I have no furtehr comments and am very much looking forward to seeing this in print

Reviewer #3: In their analysis, Procter and co-authors conducted a model-based cost-effectiveness analysis to project the health and economic consequences of a hypothetical global Group B Streptococcus maternal immunization program. This analysis was the first to incorporate the new GBD projections and synthesize outcome across 170+ countries. The analysis and manuscript are generally comprehensive, transparent and well-executed. I only have minor comments and clarifying questions. 

1. Abstract. Page 2, line 45-46, could the authors concisely describe what this range reflects? 

2. Page 6, line 171. I apologize if I missed it, but do you describe the utility decrements/QALY loss assumptions for newborn iGBS death? It is mentioned briefly later on (page 8), but could the authors describe how it's calculated, and the value?

3. Page 7, line 179. For survivors with long-term sequalae, did the authors need to assume an average age for the onset of LOGBS to determine at what age to start applying utility decrements? Similarly, for EOGBS?

4. Page 8, line 217. Small typo at the end of the line. 

5. Page 9, line 261. Could the authors briefly describe their approach to assigning distributions for the reader. In the supplementary appendix table 2, I noticed several risk parameters were assumed to be fixed, e.g., Riskmild-NDI-men. Could the authors justify? Was any correlation between parameters assumed? Is the 2.5 and 97.5 considered across all countries as well as across simulations?

6. Methods. A Bayesian disease model is mentioned in the abstract methods but is not mentioned in the main manuscript until the discussion section. Consider describing this model in some detail within the main manuscript methods. 

7. Results. Although the authors transparently report the distributions and parameterization of their uncertainty, I am left wondering what parameter(s) is(are) driving the large uncertainty bounds in the outcomes. If the bounds included uncertainty across the countries, this between-country variation may be a driver (see comment 5)). It would be informative for the reader if a regression or EVPPI analysis could be undertaken to shed light on the drivers of the uncertainty points for at least the main outcome of NMB; although, such analysis would only point to parameters that were not fixed (and descriptions would need to be set in the context of fixed vaccine efficacy). 

8. Page 13, line 319. To categorize the 103 countries, do the authors assumed a cut-point of 50% probability of being cost-effective, or are the authors using expected NMB >0 for the cutoff (preferred)? In general, I would challenge the co-authors about what additional information fig 3 adds to the messaging in order to justify inclusion, given that readers often mis-interpret the probability of an intervention being cost-effective. Although this figure reports individual-country uncertainty, the expected NMB with uncertainty bounds are reported nicely in Figure 2 (presenting the expNMB, which are used to make decisions), and I wonder if the same approach could be used for the individual-level country's NMB?

[LINK]

---

## [Decision Letter · Decision Letter 2]

3 Feb 2023

Dear Dr. Procter,

Thank you very much for re-submitting your manuscript "Maternal immunisation against Group B Streptococcus: a global analysis of health impact and cost-effectiveness" (PMEDICINE-D-22-02313R2) for review by PLOS Medicine.

I have discussed the paper with my colleagues and the academic editor and it was also seen again by 2 reviewers. I am pleased to say that provided the remaining editorial and production issues are dealt with we are planning to accept the paper for publication in the journal.

[LINK]

We look forward to receiving the revised manuscript by Feb 10 2023 11:59PM.   

Sincerely,

Philippa Dodd, MBBS MRCP PhD

PLOS Medicine

plosmedicine.org

Requests from Editors:

GENERAL

Thank you for your detailed and considerate responses to previous editor and reviewer comments which we appreciate and accept. Please see below for some further minor revisions.

FIGURES

Thank you for revising the colour palate used for your figures. Figure 1 remains unchanged, at least in my version, please revise if possible.

Previously I said the following: “Line 107 We developed a decision-tree model (Fig. 1) - please revise to “Figure. 1” Please check and amend throughout the manuscript. This was my error, please accept my sincere apologies. Figures should be labelled as “Fig” and referred to in the text as such.

DISCUSSION

Line 446: “Our study is the first…” claims of primacy can be risky, suggest “To our knowledge…” or something similar

REFERENCES

For in-text reference callouts, please ensure that citations (placed in square parentheses) precede punctuation. For example, line 448 should read “…United States [12–14], the Netherlands [17],” and line 449 should read “…37 Gavi countries in Africa [20].” 

Please check and amend throughout

SOCIAL MEDIA

To help us extend the reach of your research, please provide any Twitter handle(s) that would be appropriate to tag, including your own, your coauthors’, your institution, funder, or lab. Please detail any handles you wish to be included when we tweet this paper, in the manuscript submission form when you re-submit the manuscript.

Comments from Reviewers:

Reviewer #1: Thanks authors for their effort to improve the manuscript. I am satisfied with the response and revision. No further issues needing attention.

Reviewer #3: Thank you for your thoughtful responses. I have no further comments.

[LINK]

---

## [Editor Report · Decision Letter 3]

7 Feb 2023

Dear Dr Procter, 

On behalf of my colleagues and the Academic Editor, Professor Marie-Louise Newell, I am pleased to inform you that we have agreed to publish your manuscript "Maternal immunisation against Group B Streptococcus: a global analysis of health impact and cost-effectiveness" (PMEDICINE-D-22-02313R3) in PLOS Medicine.

Before we can publish your manuscript please ensure the following final revisions are made:

1) Line 501: please remove the data sharing statement from the main manuscript and include only in the manuscript submission form

2) Line 456: "This was the first cost-effectiveness study..." please add "to our knowledge" or "as far as we know" we previously amended the paragraph above for the same but missed this.

PRESS

Best wishes, 

Philippa Dodd, MBBS MRCP PhD 

PLOS Medicine